

# Seasonal variations in leaf and branch trace elements and the influence of a 3-yr 100% rainfall exclusion on *Pinus massoniana* Lamb

Tian Lin[1,2], Xuanmei Zheng[3] and Huaizhou Zheng[2]

[1] Fujian University of Technology, School of Ecological Environment and Urban Construction, Fuzhou, Fujian, China
[2] Fujian Normal University, Fujian Provincial Key Laboratory for Plant Eco-physiology, Fuzhou, Fujian, China
[3] Fujian Jiangxia University, Straits College of Finance and Economics, Fuzhou, Fujian, China

## ABSTRACT

**Background**. Trace elements are essential for the growth and survival of plants, and their concentrations and distributions in plants are effective reflections of ecological adaptation strategies. However, this aspect has seldom been addressed.

**Method**. Changes in the leaf and branch trace elements of *Pinus massoniana* Lamb, induced by seasonal dynamics and in response to a 3-yr 100% rainfall exclusion, were evaluated.

**Results**. The results showed that the concentrations of Fe, Cu, Zn, Cd, Ni and Cr in leaves of *P. massoniana* in the control group had high seasonal resolution. There were three groups according to their patterns over the growing season: (1) nutrient elements (Cu, Zn, Ni and Cd), which continuously decreased in concentration during the growing season, with the highest concentration in spring and the lowest in autumn; (2) accumulating element (Cr), which increased in concentration from spring to autumn; and (3) indifferent element (Fe), which increased in concentration from spring to summer and decreased in concentration from summer to autumn. The concentrations of trace elements in leaves and branches showed no significant differences with mild drought stress, except for Fe and Cr in leaves and Cr in branches, which significantly increased ($p < 0.05$) under the result of self-selection under mild drought stress. Therefore, the resultant seasonal and drought effects on trace element cycling in *P. massoniana* could provide theoretical support to respond to future climate change.

Subjects Plant Science, Soil Science, Climate Change Biology, Environmental Impacts, Forestry
Keywords Seasonal variations, Rainfall exclusion, Trace elements, *P. massoniana*

## INTRODUCTION

As an important part of the terrestrial ecosystem, the forest ecosystem has the characteristics of strong stability and perfect function (*Hisano, Searle & Chen, 2018*). Trace elements, defined as elements that are present at low concentrations (mg.kg$^{-1}$ or less) in most soils, plants and living organisms (*Phipps, 1981*), are widely distributed in the forest ecosystem and essential for the growth and survival of plants. The contamination of terrestrial environments with trace elements is a common environmental stress phenomenon

Corresponding author
Huaizhou Zheng, zhz@fjnu.edu.cn

worldwide (*Eid & Shaltout, 2014*). Fe, Cu, Zn, Cd and other trace elements in forest plants come from the soil and atmosphere, and are elements that cannot be degraded by microbial or chemical processes to alter their toxicity. Long-term accumulation of trace elements causes chronic damage to living organisms; for example, increasing the concentration of trace elements in plants affects normal growth and physiological and ecological characteristics (*Peñuelas, 2002*). Plants are sessile organisms that must cope with variable environments (*VanIeperen, 2016*), both in space and time, and this may induce changes in abiotic factors (light, temperature and water) and biotic factors (bacteria and fungi) (*Etienne et al., 2018*). For example, plants absorb heavy metals from the atmosphere and soil through respiration, and the root system potentially harbors hazards. To survive, grow and reproduce, plants must develop multiple physiological mechanisms to maintain a reasonable range of trace elements (*Philippot et al., 2013*). Many studies suggest that environmental fluctuations and features of the ecosystem (such as plant growth strategies, and soil nutrient heterogeneity) affect foliar nutrient elemental concentrations (*Han et al., 2005*; *He et al., 2015*; *Kerkhoff et al., 2005*; *Sardans, Peñuelas & Rivas-Ubach, 2011*). Changes in the concentrations of nutrient elements during plant growth can reflect the physiological and growth characteristics of plants to a certain extent. In addition, trace elements are involved in all metabolic and cellular functions, have the ability to form stable compounds with biological organic macromolecules, and can remain in plant tissue for a long time and have seasonal effects (*Baycu et al., 2006*). Therefore, the study of seasonal variations in plant trace elements can be beneficial to further understanding of the nutrient status and growth trends of the plants.

Currently, most research on the characteristics of nutrient elements is mainly focused on the concentrations of C, N, P, K and other macro elements (*Güsewell et al., 2005*; *Sardans et al., 2013*; *Urbina et al., 2015*; *Wang et al., 2013*; *Woods et al., 2003*). Because of their roles in several physiological functions in plants, such elements correlate with the growth rate (*Elser et al., 2010*; *Peñuelas et al., 2013*; *Sardans et al., 2013*) and with the structure and function of plant communities (*Elser et al., 2003*; *Sardans & Peñuelas, 2012*). The C:N:P ratio in organisms can be associated with important ecological processes, such as responses to environmental stress (*Sardans et al., 2013*; *Woods et al., 2003*), and ecosystem composition and diversity (*Güsewell et al., 2005*; *Roem & Berendse, 2000*). However, trace elements that help macronutrients improve physiological functions are not well studied (*Paiva et al., 2017*; *Sardans, Peñuelas & Ogaya, 2007*; *Waraich et al., 2011*). Some studies of trace elements vary along environmental gradients (*Eid & Shaltout, 2014*; *Paiva et al., 2017*), and their resistance to stress (*Mahdavi et al., 2016*; *Urbina et al., 2015*). Water is the most limiting factor in ecosystems and regulates plant growth and yield; a lack of water leads to a substantial number of effects on plants and species relationships, including readjustment of transport and metabolic processes, community diversity and ecosystem productivity (*Peñuelas et al., 2013*). A drought-induced decrease in soil moisture content may reduce the rate at which plants absorb trace elements. If we are to fully understand the impact of climate change on ecosystems, it is necessary to study the interactions between predicted droughts and the changes in trace elements. However, it is uncertain whether and how drought can affect trace element dynamics.

The leaf is the main site of photosynthesis and the most active organ in plant metabolism, its chemical element concentrations can reflect plant absorption and accumulation of elements (*Anderson & Proctor, 1990*; *Grubb, Turner & Burslem, 2009*), revealing the plant's demand for nutrient elements and adaptability and feedback to climate and regional changes. The branch is another important photosynthetic organ. when the leaf area is small or the leaves are degenerated, the branch adopts the role of leaves to perform physiological functions, such as photosynthesis and transpiration (*Lacointe et al., 1993*). Therefore, the concentrations of elements in branches will change with the environmental changes. Trace elements in leaves and branches show different physiological functions. For example, Fe plays an important role in regulating osmotic pressure, promoting protein synthesis, and improving photosynthetic capacity and plant resistance (*Ahanger et al., 2016*); Zn is a component of various enzymes in plants and is involved in plant respiration and carbohydrates transformation (*Blasco, Graham & Broadley, 2015*; *Hajiboland, 2012*); and Cu plays a key role in photosynthetic and respiratory electron transport chains, ethylene sensing, cell wall metabolism, and protection from oxidative stress (*Yruela, 2009*). Changes in environmental factors such as water and temperature have controversial effects on the accumulation of trace element. For example, some scholars believed that under drought conditions, the increased accumulation of trace elements can enhance the negative effects of drought on plant productivity; on the other hand, the decrease in trace element accumulation increases the productive capacity of these ecosystems and improves their capacity to resist drought (*Sardans & Peñuelas, 2007*). Other scholars have come to contrasting conclusions.

This study investigated changes in leaf and branch trace elements of *P. massoniana* induced by the annual seasons and in response to a 3-yr manipulated precipitation experiment (100% rainfall exclusion) that began in April 2013 and ended in January 2016 in Changting County, Fujian Province, China. In this experiment, the local pioneer *P. massoniana* was taken as the research object to study the seasonal dynamic of the trace elements and the influence of drought on it, to provide theoretical support for the future response of forest species to climate change. Here, the concentrations of six selected trace elements (Fe, Cu, Zn, Cd, Ni, and Cr) in the leaves and branches of *P. massoniana* growing in the control group (natural conditions) and the drought group (100% rainfall exclusion) were determined. The present research was performed to investigate (1) the seasonal dynamics characteristics of leaf and branch trace element concentrations in *P. massoniana* over the study period; (2) whether drought increases or decreases trace element concentrations and accumulations in leaves and branches of *P. massoniana*; and (3) the interaction between trace element concentrations and seasonal variations or drought.

## MATERIALS & METHODS

### Research area

The sampling site was located in Hetian Town, Changting County, Western Fujian Province, China (116°18′−116°31′E, 25°33′−25°48′N, 310 asl) (Fig. 1). This site was one of the most serious red soil erosionsites in China. Because of soil depletion, the dominant species

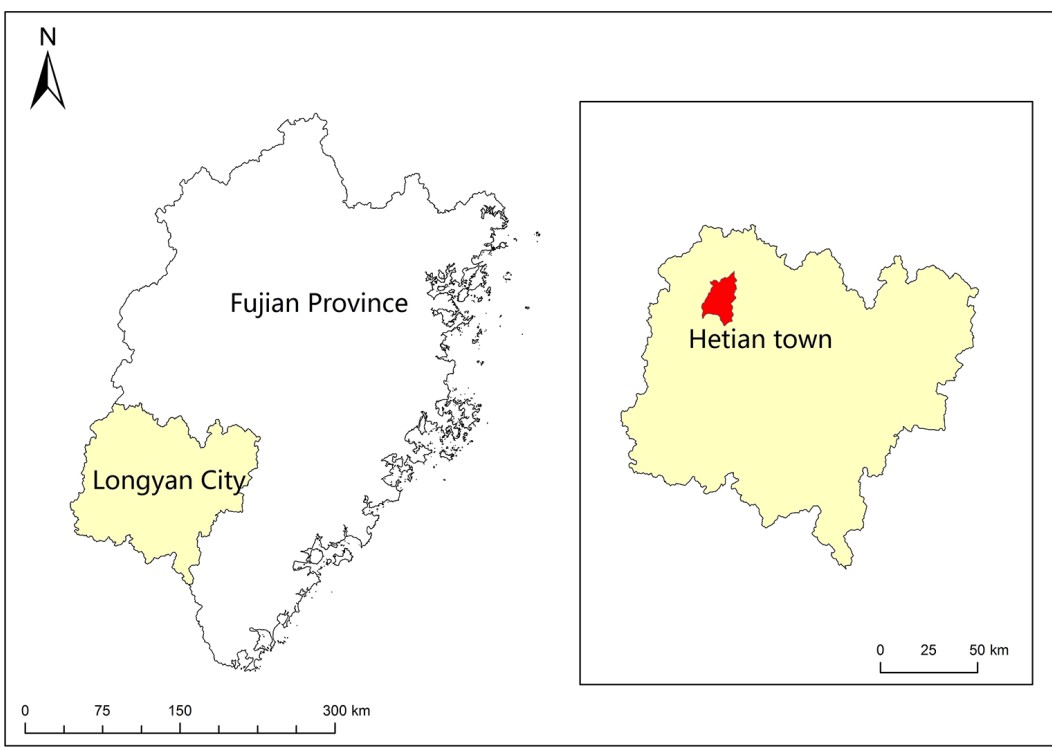

**Figure 1  Map of the study area.**

were *P. massoniana*, *Schima superba*, *Castanopsis fargesii and Lithocarpus sp.*, and there were only a few *Dicranopteris dichotoma (Thumb) Berhn* under the forest. This area had a humid monsoon climate in the middle subtropical zone, with distinct dry and wet seasons. The rainfall was mainly concentrated from May to July every year, with an average annual rainfall of 1,737 mm, which was relatively abundant. The annual average temperature in this area was 17.5 °C ∼ 18.8 °C, the lowest temperature recorded was −7.8 °C and the highest temperature recorded was 39.8 °C. An automatic meteorological station was installed to collect temperature, soil moisture, precipitation, and relative humidity measurements every 15 min at the study site. According to the linear relationship between rainfall and temperature recorded by the automatic weather station, A Bagnouls-Gaussen bioclimatic diagram (Fig. 2) was made to evaluate the drought situation under natural conditions in the study area. The method considered that when the total monthly precipitation was less than twice the mean temperature in that month defined as arid. The results in Fig. 2 show only six months with a soil water deficiency during the 34 study months (April 2013 to January 2016). Therefore, we considered that the study site without aridity could be used as a control group.

## Experimental design and sampling

The sample plot was set up on a 30° slope in April 2013 (Field experiments were approved by the farmer Ping Xiu who had the ownership of the field), which had a total 42

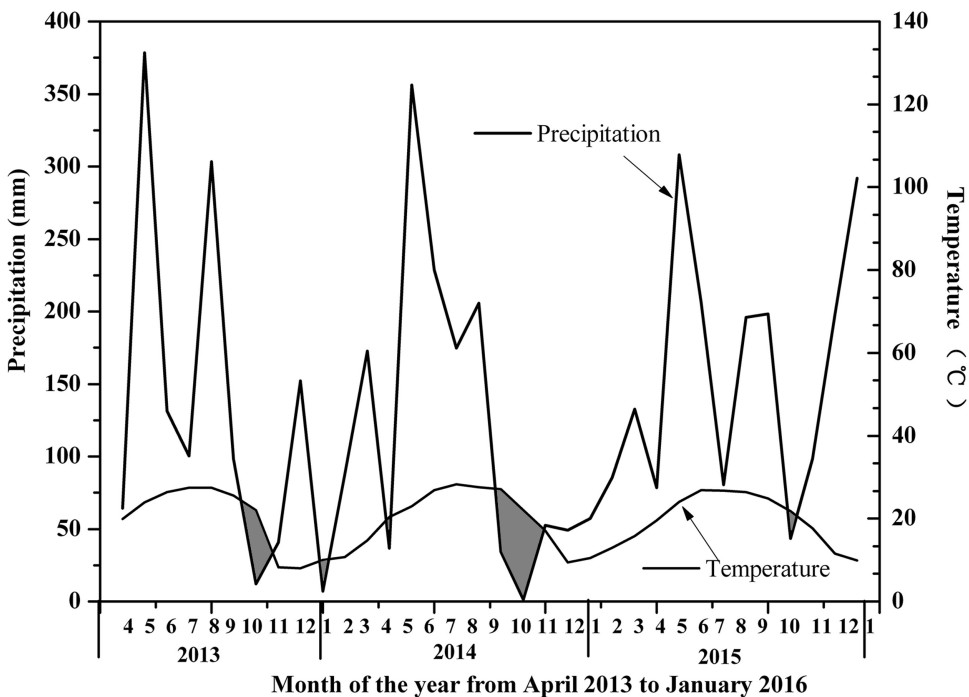

**Figure 2** Bagnouls–Gaussen bioclimatic diagram.

*P. massoniana* individuals nearly 25 years old. On average, those similar *P. massoniana* individuals were 4.2 cm in diameter at breast height (DBH), 2.4 m in height and 8.5 cm in blade length. The sample plot were divided into four small plots (20 × 20 m) (*Urbina et al., 2015*). In order to avoid the influence of slope on sampling, two small plots with 20 *P. massoniana* individuals (one uphill and one downhill) received 100% rainfall exclusion as drought plots, and the other two plots with 22 *P. massoniana* individuals in the natural environment did not receive any treatment served as control plots. In the drought group, a top rain reduction device was installed. Transparent tiles were fixed above the study plot, and all rainfall was intercepted above the whole canopy. The light transmittance of the wave pattern can reach 90%. Because the device was built only above the top of the canopy, the air convection around was sufficient, and the temperature and humidity of the two group were consistent. At the same time, to ensure the rainfall could flow out of the isolation area, the wave tiles were parallel to the plots (*Sardans, Peñuelas & Ogaya, 2007*; *Sardans et al., 2013*). Then, a soil lateral water control device was built around the study site. According to the sampling of *P. massoniana,* it was found that the main root was 2 m deep and the lateral root was less than 80 cm. Therefore, an 80 cm deep channel was dug around the study site, and aluminum plates were installed in the channel to avoid water infiltration. The control group maintained the natural state without any treatment. Finally, to prevent small animals from destroying the plots, the four plots were surrounded by wire to ensure that the test ran without interference.

On August 18, 2013 (115 days after 100% isolated rainfall), 3 *P. massoniana* individuals with relatively consistent height and diameter were randomly selected in the middle of 4 plots (total $n = 12$). The dates of plant sampling were selected based on the major climatic factors of the study site. In the study period, leaves and branches were collected from each sample individual one a quarterly basis to measure trace elemental concentrations. The 11 sampling times were as follows: day 115 (2013.8.18, late summer), day 185 (2013.10.27, autumn), day 256 (2014.1.6 winter), day 332 (2014.3.23 spring), day 467 (2014.8.1, late summer), day 542 (2014.10.15, autumn), day 638 (2014.12.30, winter), day 738 (2015.4.9, spring), day 861 (2015.8.10, late summer), day 941 (2015.10.29, autumn) and day 1032 (2016.1.28, winter) (*Anderson, 1964*).

To minimize the impact of diurnal variability, one branch with healthy leaves completely exposed to the sun was collected from the east, south, west, north directions in the middle and upper parts of the crown of the selected sample individual between 9:00 am and 11:00 am on the sampling day, and then the healthy leaves with lengths of 8.3~8.6 cm and the branches with diameters of 0.3~0.5 cm were selected to mix as one leaf sample and one branch sample, respectively. All samples were immediately stored in an ice box (0−4 °C) and brought back to the laboratory. First, the enzymatic activity of the sample was inhibited by microwave treatment at 800 W for 5 min and then the sample was dried in an oven at 65 °C for 48 h to obtain a constant weight (*Körner, 2002*). The samples were ball milled to a fine powder (Tissuelyser-24, CHN), which served to measure trace elements (*Lin et al., 2020*).

## Chemical analyses

The concentrations of Fe, Cu, Zn, Cd, Ni, and Cr were measured in leaves and branches using ICP-MS (Mass Spectroscopy with Inductively Couple Plasma, Thermo iCAP Q, USA). Before ICP-MS analyses, the samples was acid digested with an $HNO_3 \sim H_2O_2$ mixed acid system (*Martínez-Fernández et al., 2015*). First, 100 mg of dried leaf and branch powders were accurately weighed and placed in a 50 ml disposable polyperfluoroethylene (FEP) digestion tube (*Eid & Shaltout, 2014*). FEP tubes with 8 ml of nitric acid (65%) (Merck GR, Germany) and 2 ml of $H_2O_2$ (30%) were incubated overnight. The next morning, the tubes were shaking gently and placed in a graphite furnace for digestion. Gradient heating was used for the acid digestion (first step: heated to 80 °C for 10 min; second step: 100 °C for 30 min; and third step: raised to 110 °C and maintained until the white smoke was exhausted; last, the obtained solution was concentrated to approximately 1 to 2 ml). FEP tubes were taken out and cooled at normal atmospheric temperature, and then Milli-Q water was added to a constant volume. Finally, the sample solutions filtered by 0.45 μm microporous membranes were stored in the refrigerator for detection (*Malea & Kevrekidis, 2013*). Plant trace element concentrations were shown in mg.kg$^{-1}$ by dry weight.

## Method for measuring water stress

To verify the effectiveness of isolated rainfall, the leaf water potential values (Ψd) of *P. massoniana* in different treatment groups were determined within one year after the

beginning of isolated rainfall. The six target individuals (three from the uphill plot and three from the downhill plot) were chosen from each treatment, then south branches with health leaves from the upper crown of which were collected at 6:00 am on a clear day every month. They were placed immediately in a cold closet, and then taken to the laboratory to determine the leaf water potential by a WP4 dew-point potential meter (Decagon Device, Pullman, WA, USA) (*Klein et al., 2010*).

The soil moisture contents at 20 cm and 80 cm from each treatment were determined automatically at intervals of 15 min using Time Domain Reflectometry (Trime-T3 Moisture Meter, IMKO, GER) (*Sardans et al., 2013*).

### Statistical analyses

SPSS 19.0 (SPSS Inc. Chicago, USA) for Windows statistical software was used for all results. Samples from each treatment were replicated for six times. Statistical analyses were applied to obtain a distinctive view of the influence of treatment and time on the data variability. We used a one-way ANOVA to detect the effects of different seasons or treatments on the concentrations of trace elements. Repeated measures analysis of variance (ANOVA) were used to study the effects of season and treatment on the amount of trace elements. The correlation analysis was used to compare the relationships among different trace elements. Principal component analysis (PCA) was used to provide scores for observations (samples per treatment and season) and variables (trace element concentrations), which could be interpreted together. Significance and high significance levels for all tests were set at $p < 0.05$ and $p < 0.01$.

## RESULTS

### The relationship between soil moisture content and leaf water potential during a 3-yr period

Figure 3 shows the dynamic changes in soil moisture content between the control and drought treatments. From the beginning of the experiment (2013), the general trends of soil moisture content in 20 cm and 80 cm soil layers of the control group both showed a significant seasonal variation trend ($p < 0.01$) (Fig. 3). The valleys at 20 cm depth appeared in the control group on days 185 (27 October 2013, autumn) and 542 (25 October 2014, autumn), and the valleys at 80 cm depth appeared on days 185 (27 October 2013, autumn) and 638 (30 December 2014, winter). The 1,032 days of 100% rainfall exclusion caused a mean reduction in the soil moisture contents at 20 cm (mean 10.69%, CV: 4.69%) and 80 cm (mean 17.31%, CV: 10.44%) compared to the soil moisture at 20 cm (mean 22.56%, CV: 18.4%) and 80 cm (mean 30.85%, CV: 18.4%) in the control group ($p < 0.01$), decreasing by 11.87% and 13.54%, respectively (Fig. 3). The observed effects of experimental drought from 2013 to 2016 resulted in mild drought conditions in the drought group (*Hsiao & Acevedo, 1974*; *Zhang et al., 2015*), which was likely because of the movement of soil capillary water caused by the moderate slope in the sample plot.

Leaf water potential is the most sensitive indicator of water deficit and drought resistance in plant organs. In the control group, the general trend of $\Psi$d showed obvious seasonal variation. However, 100% rainfall exclusion resulted in $\Psi$d weakening continuously in

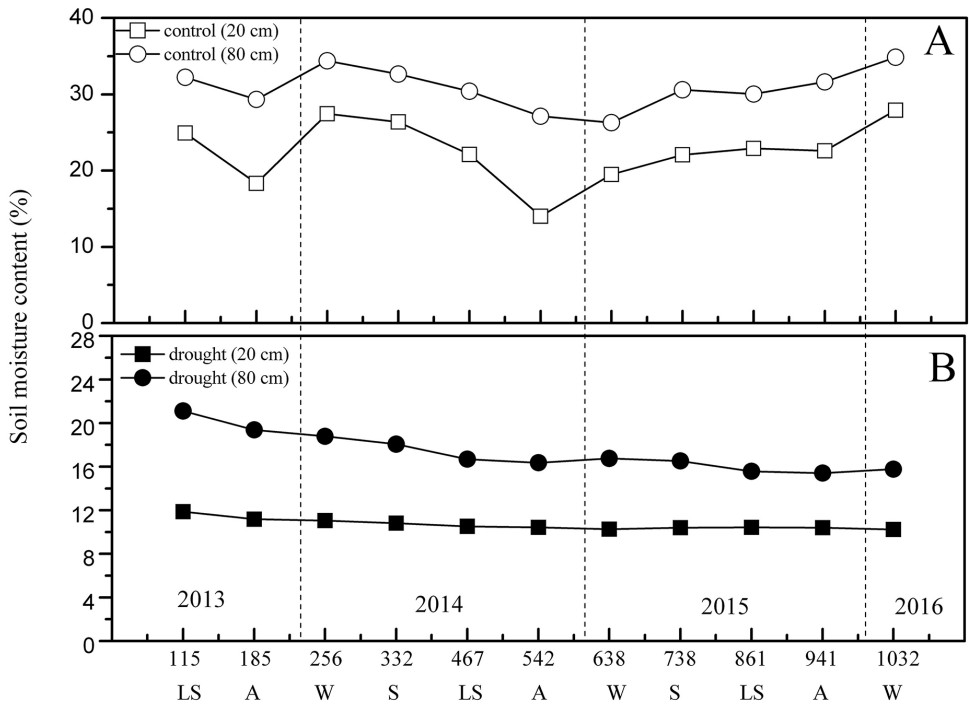

**Figure 3** **Seasonal variation of soil moisture content under different treatment in the study period.**
(A) In the control group. (B) In the drought group. S, spring; LS, late summer; A, autumn; W, winter.

the drought group from April 2013 to March 2014. The mean value in the control group
($\Psi$d: $-2.80$ MPa, CV: 10.00%) was significantly higher than that in the drought group
($\Psi$d: $-3.73$ MPa, CV: 7.78%) ($p < 0.05$) (Fig. 4). The decline in leaf water potential was
an important indicator of plant water shortage, forming a large gradient between plant
water potential and soil water potential, which was beneficial for plants to absorb water
from soil and showed that *P. massoniana* in the drought group initiated the physiological
mechanism that aims to use the lower leaf water potential to enhance the water absorption
capacity of plant cells to improve their drought resistance.

## Characteristics of trace element concentrations in plant leaves and branches

In our 3-yr experiment, the average concentrations of six trace elements in leaves of
*P. massoniana* were in 100 to 500 mg.kg$^{-1}$ (Fe), 10 to 100 mg.kg$^{-1}$ (Zn), and less than
10 mg.kg$^{-1}$ (Cu, Ni, Cr, Cd) (Table 1). The average concentrations of elements from high
to low are Fe>Zn>Cu>Cr>Ni>Cd. In branches, the average concentrations of six trace
elements were all significantly higher than those in leaves ($p < 0.01$), but except for the Cu
concentration, which was different from the leaves at 10 to 100 mg.kg$^{-1}$, and the ranges
and size sorting of other elements were the same. Compared with world terrestrial vascular
plants, the average concentrations of Fe, Ni and Cr in leaves and branches of *P. massoniana*
were all significantly higher than those of global terrestrial vascular plants ($p < 0.01$), where
Fe and Ni were nearly 2 times higher than the maximum, and Cr was 8 to 13 times higher

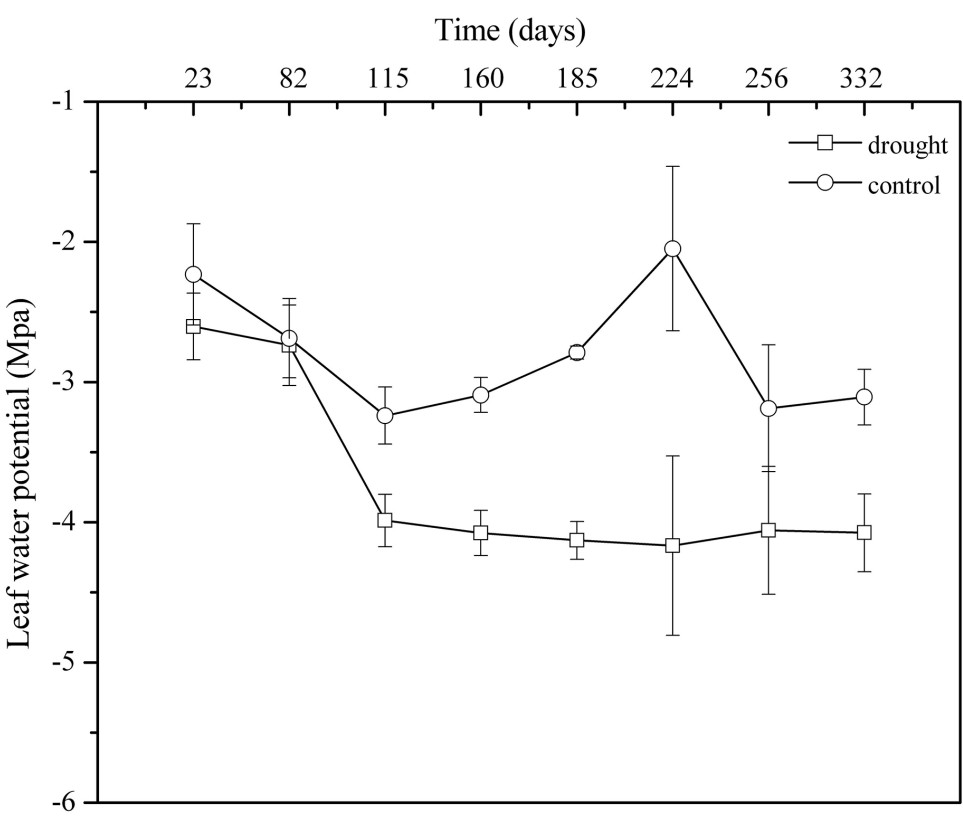

**Figure 4** Monthly patterns of dawn leaf water potentials from April 2013 to March 2014.

**Table 1** The mean trace element concentrations of *P. massoniana* across the study period in the control and drought group mg.kg⁻¹.

| Element | In the control group | | In the drought group | | Range of terrestrial vascular plant |
|---|---|---|---|---|---|
| | Leaf | Branch | Leaf | Branch | |
| Fe | 216.72 ± 14.73 | 485.93 ± 21.94 | 363.79 ± 26.09 | 496.41 ± 28.97 | 70∼180 |
| Zn | 42.25 ± 2.30 | 73.50 ± 4.47 | 46.11 ± 2.63 | 83.87 ± 3.77 | 34∼68 |
| Cu | 9.28 ± 0.53 | 15.07 ± 0.38 | 9.33 ± 0.41 | 16.12 ± 0.32 | 6∼14 |
| Ni | 2.35 ± 0.19 | 3.00 ± 0.13 | 2.64 ± 0.17 | 2.91 ± 0.22 | 0∼1.4 |
| Cd | 0.40 ± 0.02 | 1.57 ± 0.10 | 0.34 ± 0.02 | 1.34 ± 0.07 | 0.03∼10 |
| Cr | 3.07 ± 0.32 | 5.69 ± 0.22 | 3.68 ± 0.25 | 7.00 ± 0.26 | 0.1∼0.5 |

than the maximum value. The average concentrations of Zn and Cu in branches were slightly higher than that of world terrestrial vascular plants, and the remaining elements fell in the normal range. The results showed that plants in this area were at a potential risk of being poisoned by heavy metals.

## Temporal variation in trace element concentrations

An analysis of the seasonal dynamics of each trace element concentration behavior in the control group during these 3 years allowed us to distinguish three groups (Fig. 5):

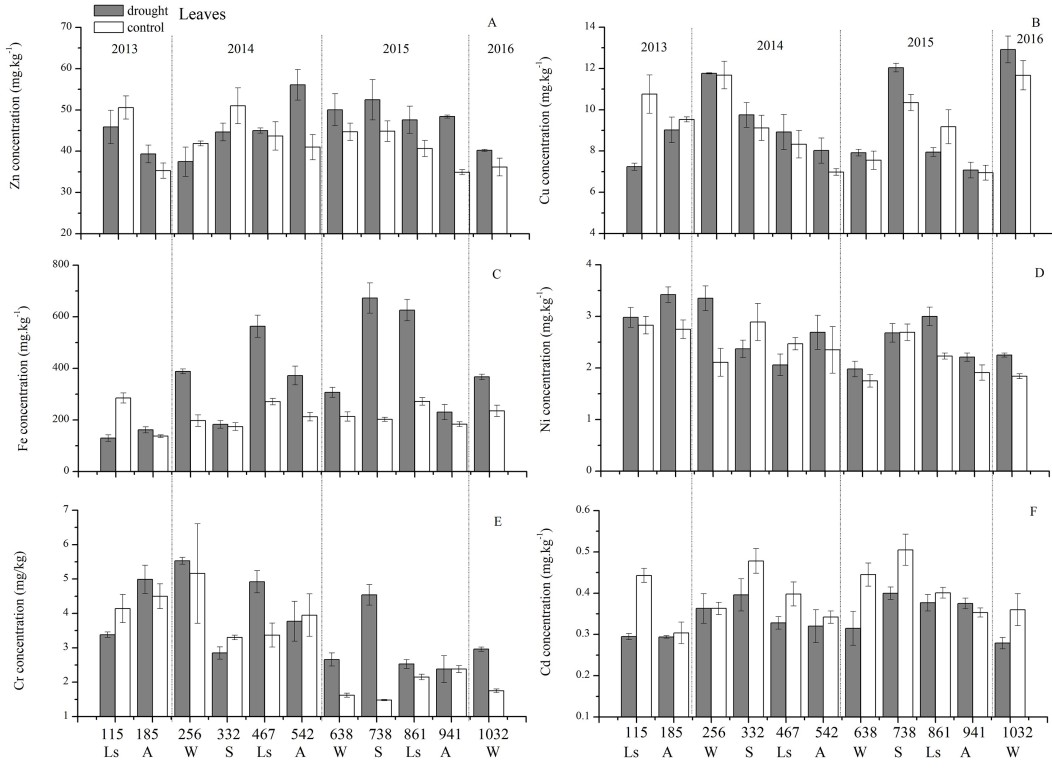

**Figure 5  Leaf trace element concentrations of Zn, Cu, Fe, Ni, Cr, Cd under different treatments (mean ± SD).** (A) Zn. (B) Cu. (C) Fe. (D) Ni. (E) Cr. (F) Cd. S, spring; Ls, late summer; A, autumn; W, winter.

(1) nutrient-like elements such as Cu, Zn, Ni, and Cd had the highest trace element concentrations at the beginning of the growing season (spring), and then decreased in concentration from spring, to summer to autumn; notably, changes were more pluralistic from autumn to winter to spring; (2) the indifferent element Fe exhibited an increase in concentration from spring to summer and then a decrease in concentration to autumn, with the highest concentration occurring in summer; and (3) the accumulating element Cr showed an increase in concentration from spring-summer to autumn, but the situations in winters were different. However, this classification was slightly different from that revealed by statistical analysis.

Figure 6A presents the results of the unconstrained PCA. PC1 × PC2 revealed two groups of variables. One group comprised Cu, Zn, Ni, Cd and Cr. The elements of this group exhibited continuous change, that is, continuing to rise or decline during the growing season (Fig. 5). The other group comprised Fe. This element first increased and then decreased. On PC1 × PC2, time scatter plots represented the position of each season category (Fig. 6B). It appeared that summers from these 3-yr study periods were significantly distinguished from other seasons (Fig. 6B). This difference could be linked to plant growth and photosynthetic processes during this active period. At the same time, spring, autumn and winter during these three years were concentrated together in Fig. 6B.

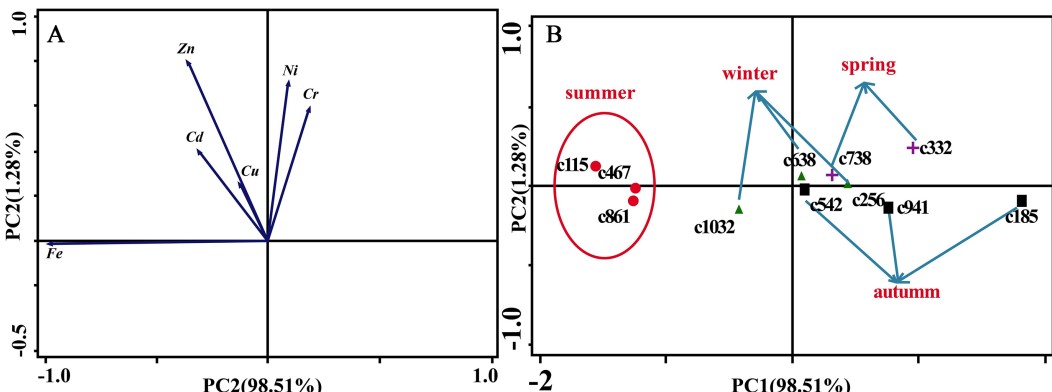

**Figure 6** **Principal component analyses of seasonal samples of *P. massoniana* leaves based on trace element concentrations in the control group.** (A) Correlation of variables. (B) Time scatter-plot of observations. C1032 = day 1032 in the control group.

Therefore, we hypothesized that trace element concentrations in leaves of *P. massoniana* in the control group collected at different seasons during the 3-yr study period could be analyzed in terms of seasonal variations.

As for branches, the Cu, Zn, Ni, Fe, Cr, and Cd concentrations of the control group changed more diversely. Although they had a certain seasonal dynamic, the annual peak appeared in different seasons during the study period (Fig. 7). An analysis of the branches revealed that all trace elements did not have a completely consistent trend with leaves of the control group (Fig. 8). With branch trace element concentrations as variables in the control group, we could not obtain useful information, and the PCA distinguished was messy (Fig. 8).

## Effect of drought on trace elements

The concentrations of trace elements in leaves and branches of the drought group showed increased concentrations of trace elements compared with those in the control group, such as Cu (0.51%), Zn (9.13%), Ni (1.23%), Fe (67.8%), and Cr (19.85%) in leaves; Cu (6.99%), Zn (14.11%), Fe (2.16%), and Cr (23.11%) in branches and decreased Cd (−14.8%) in leaves and Ni (−3.00%) and Cd (−14.5%) in branches. However, drought general resulted in no significant concentration differences under water stress conditions, except in the case of leaf Fe ($p < 0.05$), leaf Cr ($p < 0.05$) and branch Cr ($p < 0.05$). The leaf Fe concentration exhibited a clear upward trend from day 332 to day 1032, while the leaf Cr concentration increased except on days 115 and 332, and the branch Cr concentration decreased except on days 332 and 941 (Figs. 5 and 7). Figure 9 presents the results of the unconstrained PCA and shows the dates of the trace element concentrations in the leaves collected from different treatments in the 3-yr study period. The PC1 × PC2 treatment scatter-plot represents the position of each sample time in the control and drought groups. It appeared that most sample times in the drought group could be significantly distinguished from sample times in the control group. Similar processes seemed to occur in the branch tissues, and it was difficult to distinguish the sample times in the different treatments, but the 8

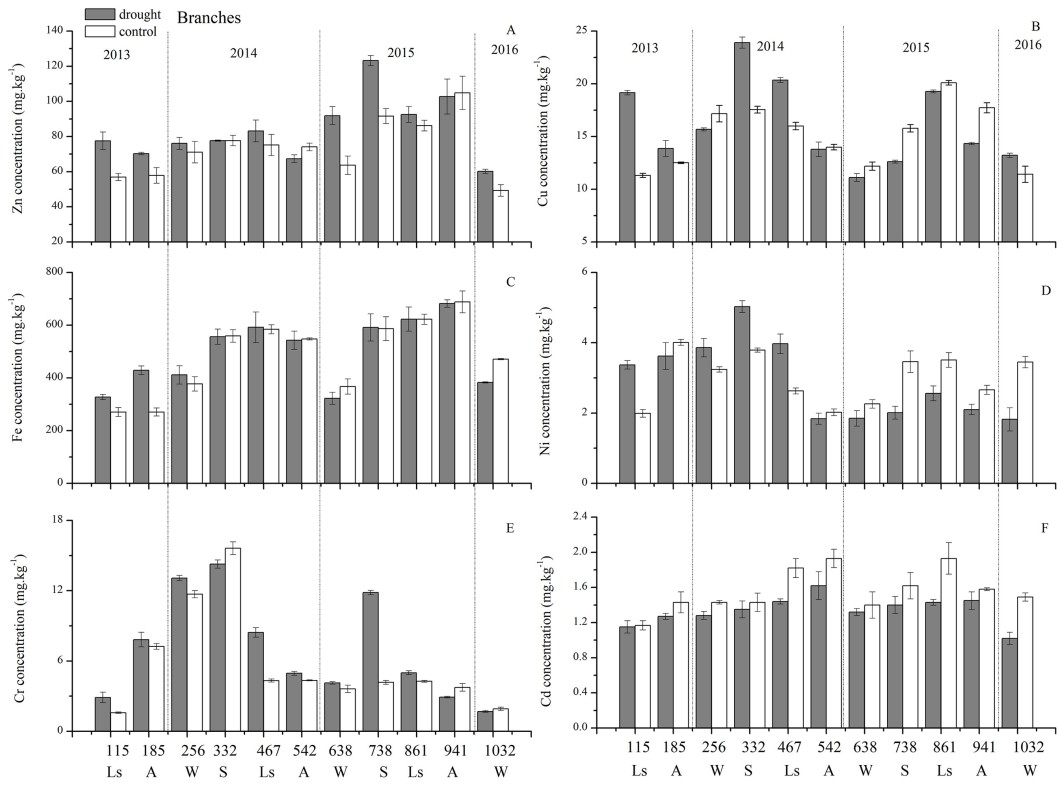

**Figure 7** **Branch trace element concentrations of Zn, Cu, Fe, Ni, Cr, Cd under different treatments (mean ± SD).** (A) Zn. (B) Cu. (C) Fe. (D) Ni. (E) Cr. (F) Cd. S, spring; Ls, late summer; A, autumn; W, winter.

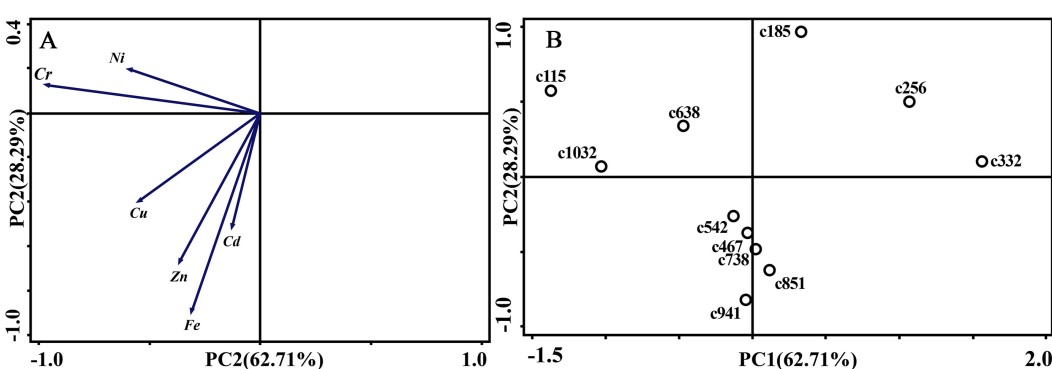

**Figure 8** **Principal component analyses of seasonal samples of *P. massoniana* branches based on trace element concentrations in the control group.** (A) Correlation of variables. (B) Time scatter-plot of observations. C1032 = day 1032 in the control group.

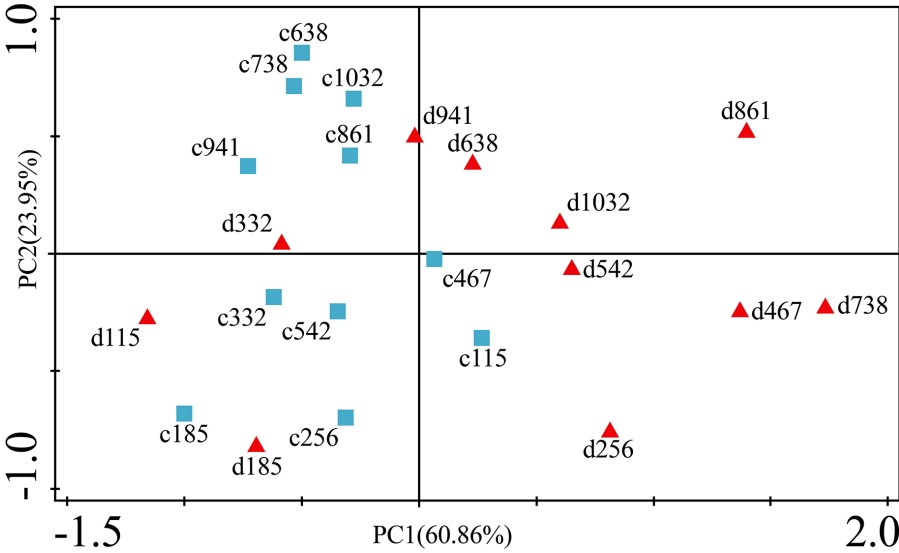

**Figure 9** **Principal component analyses of *P. massoniana* leaves based on trace element concentrations in different treatments.** PC1 × PC2 time scatter-plot of observations. c1032 = day 1032 in the control group. d1032 = day 1032 in the drought group.

groups of different treatments were clustered together in the same sampling time (Fig. 10). As a result, trace element concentrations in leaves of *P. massoniana* could be analyzed in terms of treatment variation.

## DISCUSSION

### Seasonal dynamics of trace element concentrations in *P. massoniana* in the control group

The seasonal patterns of leaf nutrient-like elements such as Cu, Zn, Ni, and Cd concentrations were similar and in agreement with the data for other plants from various environments (*Hagemeyer et al., 1992*; *Viers et al., 2012*). Leaf Cu, Zn, Ni and Cd concentrations in the control group all showed the highest in the early growth season, indicating a strong metabolic processes (high photosynthetic activity, protein synthesis energy requirements) (*Arneth et al., 1996*). Notably, from winter to spring, Cu, Zn, Ni and Cd increased. However, during these periods, the soil temperature was low and the roots could not absorb nutrients from soil (*Prokushkin et al., 2010*). Therefore, these elements probably came from the pool of elements that plant preserved in the early stage of their growth. These elements could be mobilized to the phloem before old leaf senescence and redistributed through xylem the following year to younger tissues that needs to grow (*Viers et al., 2012*). Then, the decrease in leaf Cu, Zn, Ni and Cd concentrations of the control group in the middle and late growing seasons may be explained by two processes: (1) nutrients accumulated by mature leaves were absorbed into active growing areas (e.g., fine roots and shoots) or reproductive organs; and (2) nutrients were dilution by carbon accumulation. The indifferent element Fe exhibited an increase in concentration

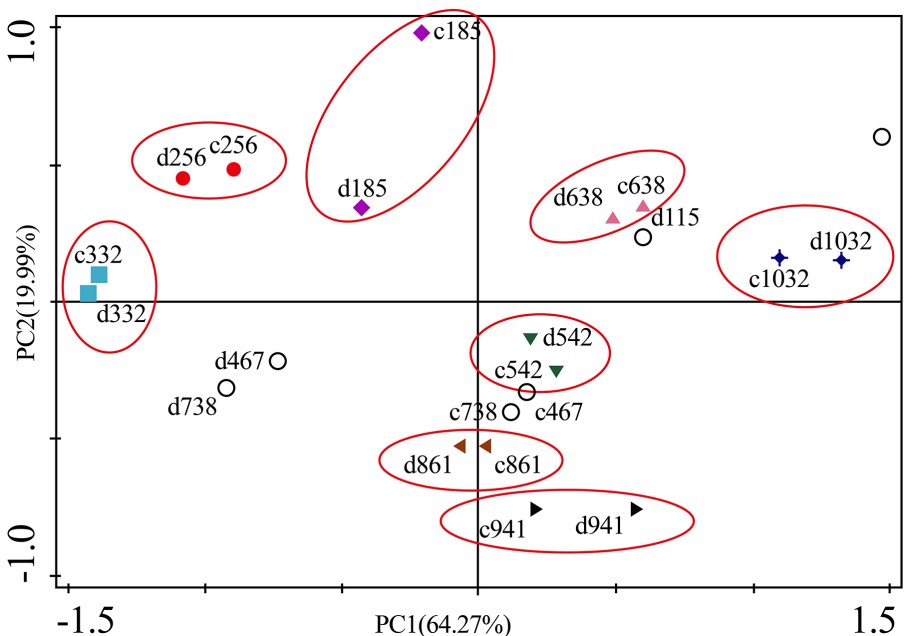

**Figure 10 Principal component analyses of *P. massoniana* branches based on trace element concentrations in different treatments.** PC1 × PC2 time scatter-plot of observations. c1032 = day 1032 in the control group. d1032 = day 1032 in the drought group.

from spring to summer and then a decrease in concentration to autumn. The highest concentration appeared in the most vigorous summer with the strongest photosynthesis. It is believed that many enzymes involved in chlorophyll biosynthesis and respiratory metabolism are Fe compounds; thus, the stronger photosynthesis was, the higher the Fe concentration. The accumulating element Cr showed an increase in concentration from spring-summer to autumn, but the situation in winter was different.

## Effects of drought on trace element concentrations in *P. massoniana* in the drought group

The 3-yr of 100% rainfall exclusion, which provoked an average reduction of 11.87% (20 cm) and 13.54% (80 cm) in the relative moisture of the soil, had some effects on trace element concentrations in the drought group. In general, water stress increased most trace elements in leaves and branches, but by statistical analysis, there were no significant concentration differences under water stress conditions except for increased leaf Fe and leaf and branch Cr concentrations significantly ($p < 0.05$). Most studies found that the increase in trace elements in the stress environment were related to the general enhancement of resource capture in soil microbe activity (*Sardans, Peñuelas & Estiarte, 2006*), and photosynthesis capacity (*Llorens, 2004*). Then enhancement of the photosynthetic capacity and the demand for resources of young leaves made the soluble organic matter migrate before senescence by the, which was the reason for the retention of trace elements in the aboveground biomass and mainly in leaves under stress conditions (*Sardans, Peñuelas & Ogaya, 2007*). In addition, the increase in trace element

concentrations in the stress environment was related to the plant showing high plant uptake, retranslocation and production capacities. Therefore, we believed that in our study site, *P. massoniana* individual as a pioneer specie of the red soil erosion area in Changting County, were able to survive easily in adverse situations due to their strong regulatory capacity. For example, *P. massoniana*, as a resprouting species, has a superficial root system, and under the water stress conditions, it could change the root structure to develop a deep root system to obtain water from the deepest layers where the impacts of drought and pollution were reduced (*Sardans, Peñuelas & Ogaya, 2007*). Even it seemed to have a dural root system (*Filella & Peñuelas, 2003*), where the deeper roots supplied moisture to the superficial roots, thereby allowing the plant to remain active and able to exploit water resources in different soil layers. In our study, the significant enhancement of the leaf Fe concentration and leaf and branch Cr concentrations in the drought group seemed to be related to an increase in the internal mobilization of resources due to the increases in photosynthetic and metabolic capacities (*Llorens, 2004*). Fe is involved in the production of chlorophyll pigment molecules. It is a component of many enzymes associated with energy transfer, nitrogen reduction and fixation and lignin formation (*Ahanger et al., 2016*). A greater leaf Fe concentration could be associated with a higher quantity of chlorophyll, which would accelerate photosynthesis in *P. massoniana* (*Sánchez-Rodríguez et al., 2010*). However, less is known about why drought could increase the Cr concentration in leaves and branches, which even enhances potential toxicity.

## CONCLUSIONS

The distribution of trace elements in plants could effectively reflect ecological adaptation strategies. The results showed that the concentrations of Fe, Cu, Zn, Cd, Ni, and Cr in leaves and branches had an obvious seasonal variation tendency in the control group, which could be divided into three types according to seasonal pattern: the first type was leaf nutrient-like elements such as Cu, Zn, Ni, and Cd, which decreased continuously with the growing season; the second type was the indifferent element Fe, which exhibited an increase in concentration from spring to summer and then a decrease from summer to autumn, and the highest concentration appeared in the most vigorous summer; the third type was the accumulating element Cr, which showed an increase in concentration from spring-summer to autumn. The concentrations of trace elements in the control group were significantly higher than those in leaves. The concentrations of trace elements in the leaves and branches of *P. massoniana* showed no significant differences with mild drought stress, except for the leaf Fe and Cr concentrations and the branch Cr concentration, which were significantly increased ($p < 0.05$). The increases in Fe and Cr concentrations were the result of self-selection by *P. massoniana* under mild drought stress.

## ACKNOWLEDGEMENTS

We thank Xiao-hui Zhong and Wan-yv Lin for their assistance in the field and lab.

### Funding

This work was supported by the National Nature Science Foundation of China (General Program: 31270659) and the Educational Research Projects for Young and Mid-aged Teachers in Fujian Province of China (Grant NO. JT180333; JAT160137; JAT190406). The funders had no role in study design, data collection and analysis, decision to publish, or preparation of the manuscript.

### Grant Disclosures

The following grant information was disclosed by the authors:
National Nature Science Foundation of China:  General Program: 31270659.
Educational Research Projects for Young and Mid-aged Teachers in Fujian Province of China: JT180333, JAT160137, JAT190406.

### Competing Interests

The authors declare there are no competing interests.

### Author Contributions

- Tian Lin conceived and designed the experiments, performed the experiments, analyzed the data, prepared figures and/or tables, authored or reviewed drafts of the paper, and approved the final draft.
- Xuanmei Zheng performed the experiments, authored or reviewed drafts of the paper, and approved the final draft.
- Huaizhou Zheng conceived and designed the experiments, authored or reviewed drafts of the paper, and approved the final draft.

### Field Study Permissions

The following information was supplied relating to field study approvals (i.e., approving body and any reference numbers):

Field experiments were approved by the farmer Ping Xiu who had the ownership of the field.

### Data Availability

The raw measurements are available as a Supplemental File.

### Supplemental Information

Supplemental information for this article can be found online at http://dx.doi.org/10.7717/peerj.9935#supplemental-information.

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
