# Peer review of "Seasonal variations in leaf and branch trace elements and the influence of a 3-yr 100% rainfall exclusion on Pinus massoniana Lamb"

_PeerJ, doi:10.7717/peerj.9935_

## Round 0.1 · original submission · Minor Revisions

Dear author I want to recognize your contribution to this important issue to improve the environment and to know the behavior of physical chemical elements of this variety of Pinus, however we consider it necessary to make minor improvements in order to strengthen the quality of your work.

Reviewer 1 ·

Basic reporting

The review overall is good, however; it needs a little more detail. Specifically where the species to work is referenced. In the body of the document the first time you quote Pinus massoniana Lamb. Put the author (Lamb.), and complete the genus).
The first letter of the genus is capitalized but the epithet is lowercase.
Check your writing well, in some sentences the subject is usually missing when it comes to It.
Try to write not so long paragraphs, example from pages 82 to 101

Experimental design

The experimental design requires support, since the author writes what was done but does not provide scientific support with works by other authors.
For the sample size, there are also works that support the methodology used (Cox, 1981).
Each measured variable says how it was done or what methodology it is used and author it

Validity of the findings

Acceptable results for a first work on this topic.
Please check the writing structure of the results carefully. Try to get the best scientific benefit. Do not describe what was found but the importance of the results

Additional comments

It is your decision whether you want to consider the opinions expressed here.

Reviewer 2 ·

Basic reporting

Clear and unambiguous, professional English used throughout
The articule was written in english and use clear and tecically correct text.

Literature references, sufficient field background/context provided.
The articule include sufficient introduction and background to demonstrate the work fits into the broader field of knowledge and appropriately referenced.

Professional article structure, figures, tables. Raw data shared.
The structure of the articule shown an acceptable format.
Self-contained with relevant results to hypotheses
Figure are relevant to the content of the article and approriately described.

Experimental design

This articule was well defined, relevant & meaningful. It is stated and fills an identified knowledge gap.
This articule shows a rigorous investigation performed to a high technical & ethical standard.
Methods used were described with sufficient detail & information to replicate.

Validity of the findings

The articule show results completaly were reached.
The articule content all underlying data have been provided; they are robust, statistically sound, & controlled..
The articule consider all conclusions and were well stated, linked to original research question to supporting results.

Additional comments

No comments related to this articule.

Reviewer 3 ·

Basic reporting

There is neither a table nor figure in the body of the article. I think it is advisable to put the matrix table of the research.

Experimental design

The experimental design is clear. In this part it is important to specify the difference between the research object and the phenomenon to be studied. The object is the pines and the phenomenon the relationship between wáter and macroelements

Validity of the findings

I think the main research question is ¿ What is the relatinship between the concentration of water and macroelements?

A general conclusión is important. ¿ Is relationship between the amount of wáter and the macroelements in leaves an branches?

Additional comments

Is a great research

---

## Round 0.2 · Minor Revisions

Dear author, we have reviewed your manuscript and we consider it necessary to make minor corrections to improve the quality of your work.

Please add citations to any literature that supports your method used in lines 194-202.

---

## Round 0.3 · accepted · Accept

Dear author, I have reviewed the minor corrections that were aimed to improve the quality of your contribution, and I confirm that you has complied according to the improvement indications. Our editorial decision is accept.